# The Impact of Polymer Electrolyte Properties on Lithium-Ion Batteries

**DOI:** 10.3390/polym14153101

**Published:** 2022-07-30

**Authors:** Nacer Badi, Azemtsop Manfo Theodore, Saleh A. Alghamdi, Hatem A. Al-Aoh, Abderrahim Lakhouit, Pramod K. Singh, Mohd Nor Faiz Norrrahim, Gaurav Nath

**Affiliations:** 1Department of Physics, Faculty of Science, University of Tabuk, Tabuk 71491, Saudi Arabia; saalghamdi@ut.edu.sa; 2Nanotechnology Research Unit, University of Tabuk, Tabuk 71491, Saudi Arabia; 3Renewable Energy & Energy Efficiency Center, University of Tabuk, Tabuk 71491, Saudi Arabia; 4Center of Excellence on Solar Cells & Renewable Energy, School of Basic Science and Research, Sharda University, Greater Noida 201310, India; 5Department of Chemistry, Faculty of Science, University of Tabuk, Tabuk 71491, Saudi Arabia; issa_hatem2@yahoo.com; 6Department of Civil Engineering, Faculty of Engineering, University of Tabuk, Tabuk 71491, Saudi Arabia; a.lakhouit@ut.edu.sa; 7Research Centre for Chemical Defence, Universiti Pertahanan Nasional Malaysia, Kuala Lumpur 57000, Malaysia; faiz@upnm.edu.my; 8Department of Materials and Earth Sciences, Technical University Darmstadt, 64289 Darmstadt, Germany

**Keywords:** electrolyte, ionic liquid, polymer, hybrid, composite, lithium-ion conductivity, lithium-ion

## Abstract

In recent decades, the enhancement of the properties of electrolytes and electrodes resulted in the development of efficient electrochemical energy storage devices. We herein reported the impact of the different polymer electrolytes in terms of physicochemical, thermal, electrical, and mechanical properties of lithium-ion batteries (LIBs). Since LIBs use many groups of electrolytes, such as liquid electrolytes, quasi-solid electrolytes, and solid electrolytes, the efficiency of the full device relies on the type of electrolyte used. A good electrolyte is the one that, when used in Li-ion batteries, exhibits high Li+ diffusion between electrodes, the lowest resistance during cycling at the interfaces, a high capacity of retention, a very good cycle-life, high thermal stability, high specific capacitance, and high energy density. The impact of various polymer electrolytes and their components has been reported in this work, which helps to understand their effect on battery performance. Although, single-electrolyte material cannot be sufficient to fulfill the requirements of a good LIB. This review is aimed to lead toward an appropriate choice of polymer electrolyte for LIBs.

## 1. Introduction

In recent decades, battery technologies have made considerable progress, mainly in LIBs. LIB demand has been increasing since 2017 from USD 29.86 billion and is expected to reach USD 139.36 billion by 2026 [1]. LIBs show very high energy density and have been widely applied in vehicles, power grids, and electrical appliances [2]. The energy density of the LIBs can commonly be improved by high-voltage active cathode materials, anode materials, and electrodes. One of the main problems regarding high-voltage cathodes and LIBs is the decomposition of the electrolyte at more than 4.2 V Li/Li+ [3,4,5,6,7,8,9]. Hence, to develop Li-ion conductivity, the stability of the material succeeding the electrolyte can also be considered a crucial electrochemical parameter in the implementation of next-generation devices that hold higher cell potential. The fast development of electric vehicles and power stations in terms of safety and energy density engenders new challenges for Li-ion batteries (LIBs) [10,11,12,13]. The progress observed in LIB technology reposes eventually on basic comprehension of the electronic, chemical and structural modifications of the battery elements, which influence the charge-discharge cycle. A LIB is made with a cathode, electrolyte, separator, and anode. The enhancement of each LIB component contributes significantly to its performance. The implementation of these batteries is highly dependent on novel electrolyte materials with outstanding transport properties, low interfacial resistance, and good mechanical strength. In lithium ion batteries, lithium ions move from the negative electrode (typically graphite) through an electrolyte to the positive electrode during discharge and back when charging. During the oxidation reaction that occurs in the process of discharging/cycling, the SEI layer fissures continuously with the mass variation of active material [14,15]. The interface stability of the active material increases the battery retention capacity during cycling and a long-time for energy storage [16,17]. The change in the content and mechanical properties of the surface of active materials has drawn particular attention [18]. For electroactive materials to be considered latent candidates for LIBs, the immediate requirement is to obtain the reversible capacity, excellent ionic conductivity, good life span, and excellent diffusion rate of lithium into active materials. The graphite (C_6_) permits the intercalation of only a single Li with the six carbon atoms that result in the stoichiometry of lithiated graphite (LiC_6_), providing the equivalent reversible specific capacity of 372 mA h g^−1^. Therefore, it is urgent to replace graphite anodes with materials having higher capacity, energy, and power density. The modification of electrolytes is a well-accepted way to improve the performance of LIBs. A small amount of electrolyte additives incorporated is often introduced to develop the SEI layer at the interface between the active material and electrolyte [19,20,21,22]. Electrolyte additive materials are often used in a small amount to create the SEI layer at the active material/electrolyte interface [23,24,25,26]. Solid-state polymer electrolytes (SPEs), a specific class of polymer, are considered promising candidates to replace current organic liquid electrolytes due to their mechanical properties and electrochemical stability against lithium metal [27]. However, current SPEs exhibit low ionic conductivity at room temperature and high contact resistance. To overcome this, there is a particular interest in gel polymer electrolytes. Gel polymer electrolytes, which contain liquid plasticizers, have received considerable attention in the field of energy storage because of their specific characteristic, high ionic conductivity, and enhanced interfacial charge transfer [28,29]. Changing the electrolyte properties is a good approach to enhancing LIB storage capacity. Figure 1 shows an illustrative schema of a LIB during the charge and discharge.

The transfer of lithium ions inside the battery is accompanied by the flow of charge in the external circuit, so the efficiency of lithium-ion transfer in the electrolyte affects the capacity of the battery.

The chemistry of secondary batteries in the aprotic electrolyte is very close to the chemistry of primary ones. There are a few recent studies that demonstrate that utilizing a liquid electrolyte in conjunction with a porous polymer structure can provide good transport properties and cycling performance in different configurations of lithium battery [31,32,33,34,35]. In the 1 M of lithium bis(trifluoromethanesulfonyl)imide (LiTFSI) with 1,3-dioxolane, Dimethoxyethane (DOL-DME) and 0.5 M LiPS (lithium polysulfide) electrolyte, the transference number was reported as 0.98, which may be the highest transference number ever reported for a gel polymer electrolyte containing free salt, along with 1.14 × 10^−3^ S cm^−1^ ionic conductivity [27]. Wang and his team used a similar crosslinked polymer as a coating on a polyethylene separator, in the presence of 1 M Lithium hexafluor phosphate (LiPF_6_) with ethylene carbonate-ethyl methyl carbonate-diethyl carbonate (EC-EMC-DEC) [34]. The transference number was 0.72 along with 10^−4^ S cm^−1^ ionic conductivity at room temperature. The impact of polyethylene dimethacrylate (PEGDMA) by the addition of water in the prepolymer solution was studied. As a result, it was found that conventional Li-ion battery electrolyte contained in the porous PEGDMA network leads to augmenting the lithium transference number to 0.79 while maintaining the ionic conductivity in the order of 10^−3^ S cm^−1^ [27].

LIBs also suffer from many drawbacks, such as the poor rechargeability of fluoromethane sulfonyl and susceptibility to misuse leading to fire or even explosion. An uncommon phenomenon of the swelling has been reported in LIBs after a complete discharge [36]. The amount of gas (hydrogen fluoride) produced upon lower voltage battery anodes is more considerable than that generated on the same electrode but with higher voltages [18]. In LIBs, the hydrogen fluoride gas generated is caused by the shortening of electrolytes on the anode after oxidation and decomposition of the SEI layer after the entire removal of lithium ions [18]. Herein, we study the impact of electrolytes on the performance of the full cell. This review focuses on recent research on various types of electrolytes for lithium battery application, mainly on investigating the effect of the additives and solvents on the electrolyte properties. In this work, IL-based polymer electrolytes, solid polymer electrolytes, gel polymer electrolytes, porous gel polymer electrolytes, aqueous polymer electrolytes, non-aqueous electrolytes, ionic conductor-based composite polymer electrolytes, and solid polymer electrolytes have been reported. This work orients in developing safer electrolytes with high voltage stability for high-energy-density LIB applications.

## 2. Base Electrolyte Salt

### 2.1. Electrolytes

The choice of a good electrolyte is one of the most important ways to resist environmental redox reactions at the electrodes and should perfectly work within the cell without decomposition. In addition, electrolytes should be inert, well-balanced, and able to operate at a certain temperature.

Carbonate-containing liquid electrolytes are mainly used to dissolve lithium salts and ease the ions’ migration within the electrolyte due to their comparatively low viscosity, but their flammability has been the main drawback that results in the use of room-temperature ionic liquids (RTILs) [37]. Ionic liquids (ILs) are non-flammable and able to operate at higher potential windows relative to carbonates and demonstrate good thermal stability with low vapor pressure. In addition, to operate, carbonates might have been inserted at a defined voltage to create an adequate SEI [37]. Fernicola and his co-workers developed and characterized a new polymer electrolyte by dissolving Li-TFSI salt in ionic liquid (IL) called: N-butyl-N-ethylpiperidinium N,N-bis(trifluoro-methane)sulfonamide (PP_24_TFSI) and the mixture was incorporated into the polymer matrix [38]. They have also studied the dynamics of atoms ^1^H, ^19^F, and ^7^Li, and their migration effect in neat ionic liquid (PP_24_TFSI), the mixture of salts (Li-TFSI: PP_24_TFSI), and Li-TFSI-doped PP_24_TFSI polymer electrolyte were investigated by the Nuclear Magnetic Resonance (NMR) method. The results indicated a significant modification of the dynamical systems due to the shifting of the minimum ignition temperature (T1) to higher temperatures. Equivalent behaviors were noticed for ^19^F and ^7^Li through the typical sequences [37]. Li-ion diffusivity in the mixed salt (Li-TFSI: PP_24_TFSI) showed a value inferior to that of PP_24_ and TFSI ions, because of the strong coordination that exists between Li^+^ ion and TFSI-anions via a coulombic interaction [37]. Nevertheless, the Li-ion transference number in the matrix was much higher, owing to the decrease in the coefficient ratios of TFSI/Li self-diffusion from 2 to about 1 moving from the salt combination to the polymer membrane. This augmentation of the transference number is greatly likable for battery applications.

The details of the methodology and defies encountered with promising electrolytes have been communicated in the literature [23,24,25,26,38]. Examples of materials reported are solid-state polymer electrolytes (SPEs) [39], with characteristically high molecular weight, gel electrolytes, or additive-based composite materials [40,41,42,43,44]. NMR spectra demonstrated a correlation between the dynamics of the polymer segment and the mobility of free charges. In this case, the results showed that the ions’ migration takes place principally in the amorphous region of the polymer at T > Tg (glass transition temperature). In addition, a rise in the diffusion coefficients of Li^+^ along the polymer chain direction was observed, which influenced the ordering of lithium cations [41]. A Li-ion transfer study was carried out in mechanically aligned polymer properties [42], or Li-ion transport has also been studied in mechanically balanced polymers [42] or polymers with balanced mechanical properties [43]. This statement is directly opposed to the ion transfer observed in polyethylene oxide (PEO), which emphasized that the migration of lithium ions occurred along the aligned axis of helical canals of the matter [44].

### 2.2. Electrolytes for LIBs

In LIBs, electrolytes are conventionally made with integrated binary carbonate solvents and contained lithium salts and exhibit low viscosity and high permittivity. Lithium salts include LiPF_6_, LiTFSI, lithium tetrafluoroborate (LiBF_4_), etc. [36]. However, these electrolytes demonstrate a low wet sensibility causing the formation of hydrogen fluoride gas [45] and the continual development of an SEI layer [46]. Organic solvents with a reduction potential of 1.0 V (vs. +/Li) are typically used in LIBs [47]. The SEI layer is formed when Li-ions, anions, and solvents interacted with each other on the surface of electrodes. However, the layer expansion affects the dendritic growth of lithium, which leads to weakening the coulombic efficiency and limits the battery life span [48]. Additionally, electrolytes made with traditional organic solvents are highly flammable, such as acetone, acetonitrile, acetic acid, benzene, acetic acid, etc., and volatile, owing to the non-stability of these organic solvents for a large temperature range [47], which can lead to the device’s degradation and conflagration. For this reason, the use of common non-organic solvents such as carbon tetrachloride, sulfur dioxide, sulfuryl chloride fluoride, dinitrogen tetroxide, antimony trichloride, and bromine trifluoride, with the potential possibilities, has led to the manufacture of non-flammable electrolytes with electrochemical and thermal stabilities.

## 3. Types of Electrolytes

### 3.1. Ionic Liquid-Based Polymer Electrolyte for LIBs

Ionic liquids (ILs) as compared to organic solvents are non-flammable with low vapor pressure, which demonstrate good characteristics to be a suitable candidate for electrolytes because of their wide potential range and their wide potential window, and excellent thermal stability. In general, ILs are classed into three main groups depending on the chemical composition of cations as shown in Figure 2 [49].

Among these classes of ILs, aprotic ILs appear to be an appropriate electrolyte material for LIBs, since they consist of ions only. However, the choice of a specific combination of anion–cation defines the IL properties, such as conductivity, viscosity, solubility, and melting point. Mostly imidazolium, pyrrolidinium, pyridinium, tetraalkylammonium, and tetraalkylphosphonium have been studied. Anions, unlike cations, were widely studied. Some examples are given in Figure 3.

The mentioned anions are inorganic anions such as halides, e.g., chloride (Cl^−^), bromide (Br^−^), iodide (I^−^), Fluorine (F^−^), polyatomic inorganic (BF_4_^−^, PF_6_^−^), or organic anions such as trifluoroacetic anhydride (CF_3_CO_2_^−^), triflate (CF_3_SO_3_^−^), bis(trifluoromethanesulfonimide (N(CF_3_SO_2_)_2_), methanesulfonate (CH_3_SO_3_^−^), and acetate (CH_3_COO^−^) [51,52,53,54,55].

Electrolytes used for LIBs are typically made with 1 M LiPF_6_ in binary carbonate solvents or 1 M LiTFSI in ternary ether solvents but suffer from stability and safety issues due to their flammability and volatility properties. To overcome this problem, Zaghib has increased the thermal stability of electrolytes using the ionic liquid 1-Ethyl-3-methylimidazolium bis(trifluoromethanesulfonyl)imide ([EMI][TFSI]) as an additive for LIBs application [56]. Chen and his team revealed that using LiTFSI and the N-methyl-N-propylpiperidinium bis(trifluoromethanesulfonyl)imide [PP_13_][TFSI] new electrolyte, crucially, enhances the rate capacity and performance at low temperatures and is more secure than using traditional electrolytes [57]. Additionally, the enhancement of the cycle stability and high-performance LIB was achieved using low-melting-point-inorganic alkali salts. Zhibin and his co-workers used dual-salt-mixed potassium bis(fluorosulfonyl)imide ([KFSI]) and lithium bis(fluorosulfonyl)imide ([LiFSI]), which showed good ionic conductivity (10^−3^~10^−2^ S cm^−1^) in a temperature range of 40−150 °C [58]. The change in IL properties via the addition of promising materials and the development of new ionic systems were reported. Three ionic liquids ([EMI][TFSI], 1-propyl-3-methylimidazolium bis(trifluoromethylsulfonyl)imide ([PMI][TFSI]) and 1-Butyl-3-methylimidazolium bis(trifluoromethylsulfonyl)imide ([BMI][TFSI]) were chosen as electrolytes and their features were studied [59]. As a result, IL ([EMI] [TFSI]), as compared to others, demonstrated the best thermal stability and electrochemical performance. Yang and his co-workers used the following five low viscosity quaternary ammonium-based ILs for LIB application: N-ethyl-N,N,N-tri-(2-methoxyethyl)ammonium bis(trifluoromethanesulfonyl)imide([N_2_(2o1)_3_][TFSI]), N-propyl-N,N,N-tri-(2-methoxyethyl)ammonium bis(trifluoromethanesulfonyl)imide ([(N_3_(2o1)_3_)][TFSI]), N-butyl-N,N,N-tri-(2-methoxyethyl)ammonium bis(trifluoromethanesulfonyl)imide ([(N_4_(2o1)_3_)][TFSI]), N-ethyl-N,N-di-(2-methoxyethyl)-N-2-ethoxyethylammonium bis(trifluoromethanesulfonyl)imide ([(N_2_(2o1)_2_(2o2)][TFSI]) and N-ethyl-N,N,N-tri-(2-methoxyethyl)ammonium bis(trifluoromethanesulfonyl)imide ([N_2_(2o1)_3_][TFSI]). Among them, ionic liquid ([N_2_(2o1)_2_(2o2)][TFSI]) and ([N_2_(2o1)_3_][TFSI]) electrolytes exhibited better properties at the current rate of 0.1 C [60]. Recently, Hirano and his team prepared a hybrid electrode using organosilicon functionalized ammonium ILs and oligo (ethylene oxide) substitute mixed with a commercial carbonate electrolyte. As a result, 30 vol% LFP-doped electrode/Li half-cell exhibited good cycle stability and reversible capacity, and efficiently drove back the decomposition of electrolyte due to the good stability of the SEI layer, thus ameliorating lithium storage performance [61]. Paillard et al. investigated the performance of the LiNi_0.5_Co_0.2_Mn_0.3_O_2_ (NCM 523)/graphite full cell. A recent study reported that 1-Butyl-1-methylpyrrolidinium ([Pyr14]), 1-Methyl-1-propylpyrrolidinium ([Pyr13]), and N-methyl-N-pentylpyrrolidinium ([Pyr15]) have high ionic conductivity and superior electrochemical performance, which could be used for LIB application [62,63,64]. The results revealed that 1-Butyl-1-methylpyrrolidinium bis(trifluoromethanesulfonyl)imide ([Pyr14][TFSI]), 1-Butyl-1-Methylpyrrolidinium Dicyanamide ([Pyr14][DCA]) and N-butyl- N-methyl pyrrolidinium trifluoromethanesulfonyl-N-cyanoamide ([Pyr14][TFSAM]) showed better electrochemical performances than electrolytes based on organic carbonate solvent in LiNi_0.5_Co_0.2_Mn_0.3_O_2_ (NCM 523)/graphite full cells [65]. An emerging way to achieve high oxidation stability (>5.5 V vs. Li^+^/Li) for a graphite and Li metal anode was performed by Qian et al. 4 M LiTFSI salt was added to IL:1-Methyl-1-propyl-3-fluoropyrrolidinium and bis(fluorosulfonyl)-imide ([PfMpyr][FSI]), resulting in a higher potential window [66]. Howlett et al. [67] manufactured a high-energy-density LIB using LiFSI salt and N-propyl-N-methylpyrrolidinium bis(fluorosulfonyl)imide anion ([C_3_mpyr][FSI]) that eliminated dendritic growth despite the rapid rate of charging. In terms of stability, although IL-doped electrolytes show lower ionic conductivity than traditional carbonate electrolytes (8–12 S cm^−1^), the entire performance of IL electrolyte-based LIB remains almost constant and shows rather higher stability. However, the full cell operated at high current densities, which enhanced the coulombic efficiency of 0.96 at 20 mA cm^−2^ with a 0.2 V polarization [47]. The morphology structure revealed the growth of uniform sediment without dendrites with low electrode resistance, as shown in Figure 4.

### 3.2. Solid Polymer Electrolytes

Solid lithium batteries can be developed using two types of electrolytes: (a) Inorganic ionic conductors or (b) solid polymer electrolytes. It has been reported that solid lithium battery In–Li_x_/Li_1−x_CoO_2_ was made using L_i3_PO_4_-Li_2_S-SiS_2_ glass-ceramic electrolyte. As a result, the charging and discharging behavior remain constant over 80 cycles, with excellent cyclic behavior, and the cell can operate under high-pressure circumstances [69]. Figure 5 shows the entire performance of a 10 Wh LiCoO_2_-based solid LIB made with a solid polymer electrolyte and put on test at 60 °C. The cell shows a good cycling life at 70% DOD and no real change was observed in the discharge voltage after 355 cycles without capacity loss [70].

In 1991, Armand reported a Li-ion-conducting electrolyte made with novel LiTFSI salt as an ion lithium-ion conductive electrolyte for LIBs and later on a new type of single-ion solid polymer [71]. Jungdon Suk et al. [72] developed highly conductive electrolytes with an ionic conductivity of 7.6 × 10^−4^ S cm^−1^ for solid-state LIBs. The polymer electrolyte film was made with lithium-salt via the in situ radical polymerization of a precursor solution containing Li-salt and polyethylene glycol dimethyl ether as a plasticizer, and a combination of pentaerythritol tetrakis (3-mercapto propionate) and a synthesized hexakis(allyloxy)cyclotriphosphazene (thiol-ene PAL) employed as cross-linker. They assembled the battery using lithium iron phosphate (L_i_FePO_4_) as the positive electrode, solid polymer electrolytes (SPE), and lithium foil as the anode. The cell exhibits good performance, the capacity of initial discharge was 147 mAh g^−1^ at 0.1 C and 132 mAh g^−1^ at 0.5 C, and 97% of the capacity of retention was found to be 0.97 at the 100th cycle [72]. The SPE-based LIB demonstrated a stability potential window of 5.66V and good mechanical stability. Figure 6 shows the entire performance of a L_i_FePO_4_-based solid LIB made with a solid polymer electrolyte.

### 3.3. Gel Polymer Electrolytes

#### 3.3.1. Gel Polymer Electrolytes for LIBs

A gel polymer electrolyte (GPE) is formed when an organic electrolyte is trapped in the porous structure. A gel-based battery is fabricated by transforming the H_2_SO_4_ aqueous solution into a gel. Gel-LIBs show different benefits such as high flexibility, lower interface resistance, better behavior at low temperature, no leakage, better safety, lower self-discharge, good working performance, higher reliability, longer cycling life, and a higher energy density. It should be noted that these characteristics mentioned above were not the same at the initial level but increased progressively until they achieved this stage of performance. As an example, Ultralife Co. Ltd. [73] assembled gel lithium batteries, which exhibit good performance as shown in Figure 7. Gel polymer electrolyte was prepared by addition of poly (vinylidene fluoride-hexafluoropropylene (PVDF-HFP) copolymer and 1 M LiPF_6_ solution in ethylene carbonate/dimethyl carbonate (EC: DEC) (*v*/*v* = 2:1). The cell performance indicates a good cycling life and better durability to overcharge than ordinary lithium-ion batteries [74]. In general, gel-LIBs operate better than batteries based on liquid electrolytes.

#### 3.3.2. Porous Polymer Gel Electrolytes

Boz and his team [27] developed new porous polymer gel electrolytes for LiB application. The electrolyte of O-PEGDMA-VS-0 was prepared by trapping the conventional electrolyte (1 M LiPF_6_ in EC-DEC) in a microporous polymer network. The prepared porous polymer gel electrolytes show a conductivity of 10^−3^ S cm^−1^ and the cation transference number rises to 0.79 compared to non-porous. Condensed polymer electrolyte developed with an equal tendency shows lower conductivity and a transference number of 7.6 × 10^−4^ S cm^−1^ and 0.65, respectively [74]. The incorporation of the porous electrolyte within Li-metal/LiFePO_4_-based cells showed an increase in its rate capability, the capacity of retention, and the efficiency of the cell caused by the enhanced ion transfer properties of the porous polymer electrolyte compared to a commercial separator [74].

Zhang et al. developed a PVDF-HFP-based gel polymer membrane with multi-sized honeycomb-like porous architectures for lithium batteries beneficial to ameliorated safety [75]. The manufactured porous membrane demonstrates high porosity (78%) that drives to a high electrolyte uptake of 86.2%, exhibits an ionic conductivity of 1.03 × 10^−3^ S cm^−1^ at room temperature, and is better than using a commercial membrane [75]. The capacity of the full cell was 63.1 mAh g^−1^ at 5 C higher than that obtained with thinner, and conventional separators.

### 3.4. Aqueous Polymer Electrolytes

Wang and his co-workers manufactured solid-state aqueous polymer electrolytes by healing water-in-salt electrolytes (WiSE) into the network of poly (methyl methacrylate) (PMMA). The prepared solid-state aqueous polymer electrolyte (SAPE) showed higher ionic conductivity compared with traditional solid polymer electrolytes [76]. The use of WiSE contributes significantly to Li-ion transport compared to that of SPE using ionic liquids as reported recently. The aqueous SAPE can resist cutting and function in an open cell condition due to its strong bonding, which maintains the water and salt in the framework of the polymer [63]. They placed a water-free thin PEO-LiTFSI-KOH SPE interface between the anode and electrolyte to further increase the current efficiency. After a test, the LiMn_2_O_4_/Li_4_Ti_5_O_12_ full cell based on SAPE@SPE electrolyte offers a practical capacity ratio of 1.14 and 12m with unparalleled high initial coulombic efficiency of 90.50%, and the average value was found to be 99.97% at 0.5 C [76]. It was found that the energy of the entire cell can be ameliorated by extending the area capacity of the active material up to 1.5 mAh cm^−1^, and the potential window of aqueous electrolytes showed an extension from 3.0 V of 21 m WiSE to ~3.86 V of 12 m UV-cured SAPE [76]. However, solid-state aqueous polymer electrolytes offer a good opportunity to overcome the safety problems of LIBs and enable them to attain high voltage and high energy modules of the battery [76]. Zhao and co-workers [77] studied a new cell—LiFePO_4_/C (cathode) |9 M LiNO_3_ aqueous solution|LiV_3_O_8_ (anode)—to achieve a higher rate capability and better cyclic stability. It showed a capacity of 88.7 mAh g^−1^ after 100 cycles at a rate of 10 C and a discharge capacity of 60 mAh g^−1^ after 500 cycles at 50 C. Lithium sulfate (Li_2_SO_4_) as an aqueous electrolyte was used in the cell: LiTi_2_ (PO_4_)_3_ (cathode)|0.5 M Li_2_SO_4_ aqueous solution|LiFePO_4_ (anode). It showed a capacity of retention of over 90% up to 1000 cycles. This aqueous electrolyte was used to neaten the PH values of the electrolyte by oxygen elimination [65]. A complete LIB of 2.3 V based on water-in-salt electrolytes (aqueous solutions) was tested to have a coulombic efficiency of nearly 100% with several cycles > 1000, which could perfectly challenge LIBs made with non-aqueous solutions in terms of energy and power density (Figure 8).

### 3.5. Non-Aqueous Electrolytes

Since 1990, commercial LIBs have been dominated by the use of a non-aqueous solution that consists of salt (LiPF_6_) dissolved in organic carbonates, especially the mixture of EC with dimethyl carbonate (DMC), propylene carbonate (PC), ethyl methyl carbonate (EMC), and/or DEC [78,79]. However, new organic solvents and Li salts are needed for LIB performance enhancement. In general, conventional organic carbonate solvents exhibit a very low potential stability, and this drives the development of electrolytes with a wide potential window [80]. This instability was observed by Lucht et al. [81] when they studied the interfacial properties between an electrolyte prepared with 1M LiPF_6_ dissolved in EC:DMC:DEC at equal proportions (*v*/*v*/*v* = 1:1:1) and LiNi_0.5_Mn_1.5_O_4_-based electrodes. The common electrolyte appears to be unstable when the charge exceeds 4.5V after testing at different potentials [4–5.3 V vs. Li].

For this reason, organic fluoro-compounds have been employed as high-voltage organic solvents since the fluorinated molecules exhibit higher electrochemical stability potential windows because of the intense electron-removing effect of fluorine atoms [80]. Zhang et al. [82] studied the electrochemical stability of various fluorinate-based electrolytes at high-voltage conditions, and the E5 electrolytes made with the composition (1.2 M LiPF6 in F-AEC/F-ethyl methyl carbonate (EMC)/F-EPE (*v*/*v*/*v* = 2:6:2)) showed a higher potential window than that of the Gen2 electrolyte based on EC/EMC. In addition, the absence of EMC/F-EMC and EC/F-AEC has ameliorated the maximal potential as shown in Figure 9.

Solvents have a crucial impact on the fabrication of a good electrolyte. The physical and chemical properties of some commonly used solvents are shown in Table 1.

Chen and his co-workers [82] developed a new non-aqueous fluorinated lithium borate cluster salt (Li_2_B_12_F_12x_H_x_) electrolyte with lithium difluoro(oxalate)borate (LiDFOB) as the additive, which showed higher thermal stability and better cycle life than LiPF_6_-electrolyte. The novel Li_2_B_12_F_9_H_3_ electrolyte showed a capacity of retention of 70% with 1200 cycles under 55 °C. Li et al. [83] used a mixture of LiPF6 (1 M) dissolved in DMC, EMC, and EC (1:1:1) as electrolytes with a lithium manganite (Li_2_MnO_3_)-doped Li_2_Ni_0.3_Co_0.3_ Mn_0.3_O_2_ cathode material. The Li_2_MnO_3_-based electrolyte ameliorated the cycling life and demonstrated a high coefficient of Li^+^ diffusion and good rate capacity.

### 3.6. Ionic Conductor-Based Composite Polymer Electrolytes

#### 3.6.1. Mechanism of Li-ion Transport

Inorganic solid electrolytes have received attention due to their good properties, such as non-flammability and high ionic conductivity up to 10^−2^ S cm^−1^ at room temperature (RT) [84,85], but showing a poor interaction with electrodes which has limited their use in commercial LIBs [86]. To overcome this inconvenience, many types of research have been conducted to enhance the interfacial properties of inorganic solid electrolytes by doping inorganic solid electrolytes used as fillers to solid polymer electrolytes (SPEs) to combine both the properties of inorganic polymer electrolytes (good conductivity, strength) and SPEs (flexibility and good interfacial properties) [87]. Polymer is not only expected to dissolve Li-salt but also be enabled to connect with Li^+^ ions. In this case, the polar groups (— O —, — S —, etc.) present in the polymer are essential for Li-salt dissolution. For this reason, researchers have been directed to PEO polymer and its offshoots. In PEO, the oxygen atom (—O—) present on the polymer chain, and lithium coordinate with each other by coulombic interactions. Under the electric field, Li^+^ cations move from one coordination point to another along the polymer segment. The mechanism of lithium-ion transfer in PEO-doped polymer electrolytes is shown in Figure 10 [79].

In the PEO-based composite system, Li^+^ ion transport is influenced by the large size of the polymer segment and the bound effect of the crystalline regions. However, lithium-ion conductivity relies on the ion’s density and the polymer chain mobility. The polymer aptitude to dissolve lithium salt defines the number of ions that can move within the polymer matrix, and thus polymer with high dielectric constant and low lattice energy lithium salt favors the dissociation [88]. Under normal conditions, the ionic conductivity (σ) can be calculated using the following expression [89]:(1)$$\sigma =F\overset{k}{\underset{j=0}{\sum }}n_{j}q_{j}{µ}_{j}$$

Here, σ is expressed in S m^−1^, F  is the Faraday constant in C mol^−1^, $n_{j}$ is the carrier density in m^−3^, $q_{j}\;$ is the number of charges in C, and ${µ}_{j\;}$ is the mobility in m^2^ (V⋅s)^−1^. It appears clearly that the increase in density of free charges and the speed of migrated ions can improve the ionic conductivity in polymer electrolytes.

The motion of ions in SPEs can be explained by three theories: Arrhenius theory, Vogel–Tammann–Fulcher (VTF), William–Landel–Ferry (WLF), and combining all these theories [90].

Arrhenius established the relationship between temperature and ion migration caused by polymer chain displacement as [91]:(2)$$\sigma =\sigma_{0\;}\exp\left(\frac{-E_{a}}{KT}\right)$$

Here, $\sigma_{0\;}$ represents the pre-exponential factor in S m^−1^, $E_{a}$ is the activation energy in J, *K* is the Boltzmann constant in J/*K* and T is the thermodynamic temperature in *K*.

The conductivity is generally affected by the mobility of the polymer segment or/and the relaxation of the polymer chain and jump motion of ions along the polymer segment, where σ = f (T) is a nonlinear curve [92]. VTF describes the relation between σ of polymer electrolyte and *T* [91].
(3)$$\sigma =\sigma_{0\;}T^{-1/2}\exp\left(\frac{-A}{T-T_{0}}\right)$$

Here, A is an action factor (with energy dimension) in J/*K*, and $T_{0}$ is the reference temperature in *K*, normally found in the range [10–50 °*K*] below the experimental glass transition temperature ($\;T_{g})$. At ambient temperature, low $T_{g}$ could have a significant impact on the conductivity enhancement if only the impact of the polymer segment on σ was taken into account.

William–Landel–Ferry (WLF) established a relationship between conductivity, temperature, and frequency. The WLF theory takes into account the relaxation process of molecular chain motion in an amorphous system. The WLF equation is:(4)$$l_{g}\left(\frac{\sigma \left(T\right)}{\sigma \left(T\right)}\right)=\frac{B_{1}\left(T-T_{g}\right)}{B_{2}+\left(T-T_{g}\right)}$$

Here, *σ*($T_{g}$) is the ionic conductivity in S m^−1^ at$\;T_{g}\left(K\right),\;$and $B_{1}\;$ and $B_{2}$ are the WLF free volume parameters.

$T_{g}$ is a crucial parameter in the conductivity improvement in the polymer electrolyte. The conductivity is very low if $T<T_{g}$ and increases undeniably if$\;T>T_{g}$. However, a reduction in $T_{g}\;$ leads to an increase in the ionic conductivity.

The mechanism of ion transport of PEO-doped polymer electrolytes has been well-explained by the three above-mentioned theories. The amorphous regions of the polymer contribute efficiently to the diffusion of ions. This theory is also applicable to other polymer-based solid electrolytes.

Figure 11 shows various composites that were doped into polymers, including fast-ion conductive ceramics [92,93,94,95], inert ceramic fillers [92,96], lithium salts [97], ionic liquid [98], etc.

#### 3.6.2. Electrolyte/Electrode Interface

A solid lithium-ion battery is typically made of LiFePO_4_ or LiCoO_2_ (cathode) and lithium metal (anode). Each electrode/electrolyte contact should have good properties that help the battery to be working well. Cathode/electrolyte contact desires a highly flexible and solid polymer electrolyte to reduce the interface resistance while the electrolyte/anode contact desires a solid electrolyte to resist the lithium dendrite growth [100,101,102]. High flexible solid polymer electrolyte allows for lowering the resistance at the interface, but poor mechanical stability could not confront the perforation caused by metallic lithium dendrites. To overcome this problem, an inorganic ceramic electrolyte with good mechanical properties is employed to sustain the metallic lithium dendrites but still suffers from poor interface contact with the electrodes (large interface resistance) [103]. Additionally, the non-stability at the interface creates side reactions and the SEI layer is formed on the electrode surface, which may conduct to reduced battery cycle life [103]. To consider the beneficial properties of both polymer and inorganic ceramic electrolyte in terms of good interface contact and excellent mechanical properties, respectively, polymer composite inorganic ceramic electrolyte has been developed [99].

#### 3.6.3. Categories of Polymer-Based Composite Electrolytes for Lithium Batteries

##### Inert-Oxide-Ceramic-Based Solid Polymer Electrolytes

To improve the mechanical features, reduce the degree of crystallinity in polymer and improve the ionic conductivity of SPE, inert oxide ceramics have been used in polymer electrolytes. Weston and his colleague [104] ameliorated ionic conductivity and mechanical properties of solid polymer composite electrolytes using PEO-doped with Al_2_O_3_ as inert oxide ceramic particles. Tambelli and his co-workers [105] revealed that the incorporation of Al_2_O_3_ into PEO can also significantly diminish the crystallinity and $T_{g}$ of PEO. This indicates that the reduction in crystallinity in the polymer leads to improving its ionic conductivity. This decrease in crystallinity can then allow a high number of free polymer segments and speed up the segmental motion that facilitates the lithium diffusion. Liang and his team [106] reported the preparation of PEO-PMMA-LiTFSI–Al_2_O_3_ composite electrolyte using the solution casting technique. The addition of Al_2_O_3_ as filler improved the ionic conductivity from 6.71 × 10^−7^ to 9.39 × 10^−7^ S cm^−1^. Lee et al. developed SPE using PEO-EC-PC and SiO_2_ as inert ceramic fillers. The conductivity of the SPE was found to be 2 × 10^−4^ S cm^−1^ at RT with a composition of 2.5wt% SiO_2_ [107]. Lin et al. [108] developed a new PEO-silica aerogel composite polymer electrolyte (CPE), and SPE shows a high Young’s modulus of 0.43 GPa and high ionic conductivity of 6 × 10^−4^ S/cm. The investigation shows that the dispersion of SiO_2_ powder into polymer has enhanced the mechanical properties and ionic conductivity. Pal and his group [109] prepared PMMA-LiClO_4_-TiO_2_-based SPEs via the solution casting method, the study showed that incorporation of composite nanosized TiO_2_ into SPE exhibits ionic conductivity of 3 × 10^−4^ S cm^−1^ at RT, and this can ameliorate the thermal stability as well. LiCoO_2_-based SPE/graphite demonstrated a specific capacity of 30 mAh g^−1^ in twenty cycles at ambient temperature. In recent research work, Tao and his team [110] fabricated PEO-LiTFSI-based SPE using Mg_2_B_2_O_5_ nanowires. The prepared composite polymer electrolyte showed excellent mechanical properties and remarkable potential windows. CPE shows good ionic conductivity due to the facility of ions to migrate upon the surface of Mg_2_B_2_O_5_ nanowires, and also due to the high-speed mobility of Mg_2_B_2_O_5_ and the coordination of Mg_2_B_2_O_5_ with TFSI^−^ (Figure 12).

##### MOF-Based Composite Electrolytes

Metal-organic frameworks (MOFs) are new porous materials containing bridging organic ligands and metal ions [111,112,113,114]. MOFs are widely employed in different domains, including molecular separation, gas adsorption, and drug delivery [115,116,117], because of their porosity, high surface area, and polymetallic sites [118]. Yuna and his co-workers [118] developed a new composite polymer electrolyte using Zn-based MOF-5, LiTFSI, and PEO polymer. The addition of MOFs to the polymer host (PEO) enhanced the mechanical and electrochemical properties of SPEs. The enhancement of ionic conductivity up to 3.16 × 10^−5^ S cm^−1^ at 25 °C may be caused by: (a) the restraint of the PEO crystallization due to the interaction between the polymer chain with the Lewis acidic sites on the MOF-5 and the lithium salt, which eases the formation of Li^+^ ion conductive channels, and (b) the adsorption of solvent through the MOF pores, which can speed up the migration of lithium ions. Gerbaldi et al. [119] used Al-based MOF as new filler material in the PEO matrix. Composite polymer electrolytes (CPEs) showed an ionic conductivity of two orders of magnitude higher than non-mixed MOFs. The composite film was utilized in LIB (Li/LiFePO_4_) as the electrolyte, and the composite-based cell demonstrated a high specific capacitance and remarkable electrochemical performance.

##### Cellulose-Based Solid Polymer Electrolytes

Cellulose is used in polymer electrolytes because it possesses good mechanical properties with a large specific surface area, it is completely safe and cheap, and it uses environmentally friendly materials [120,121,122]. The incorporation of cellulose into the polymer matrix will not only improve the mechanical properties of polymer electrolytes but also acts physically as a block to stop the formation of Li dendrites. The polar groups in cellulose can also facilitate salt dissociation in the polymer matrix [123]. During the interaction, the cellulose/polymer separation forms a channel that eases the migration of ions. Nair and his co-workers [124] prepared cellulose-based polymer electrolytes to enhance the electrical and mechanical properties of solid electrolytes. The cellulose-based electrolytes exhibit ionic conductivity of 2.0 × 10^−4^ S cm^−1^ at RT and good flexibility, which can be appropriate for solid flexible LIBs. Mixing an ionic liquid with cellulose is also an innovative solution to eliminate the leakage problem in IL-composite electrolytes. Asghar et al. [125] designed quasi-solid polymer electrolytes using network cellulose (NC) in PEG-LiCloO_4_. The solid electrolyte doped with 12.8 wt% NC indicates ionic conductivity of 10^−4^ S cm^−1^ at RT with a decomposition voltage of up to 4.7 V.

##### Garnet-Type Composite Polymer Electrolytes

Wu et al. [126] reported that a garnet-type-based lithium solid–polymer electrolyte shows a large electrochemical potential window and high ionic conductivity. However, the implementation of garnet-type ceramics into the entire battery always exhibits low conductivity at the contact between electrolyte and electrode, due to increased interfacial resistance and low ionic conductivity leading to lowered battery performance [127]. However, the combination of polymer and garnet in composite electrolytes contributes to the overall electrochemical performance of the cell. A nanoscale garnet ceramic filler with a great specific surface area enhances the ionic transition rate [128], which increases the ionic conductivity. A composite electrolyte was prepared using PEO and Li_7_La_3_ Zr_2_O_12_ (LLZO) particles doped at 52 wt.%, and the ionic conductivity of CPE was 4.42 × 10^−4^ S cm^−1^ at 55 °C [129]. Hybrid electrolytes were developed with Li_6.75_L_a3_Zr_1.75_T_a0.25_O_12_ (LLZTO) ceramic filler in PVDF [130]. It was reported that a 10 wt.% LLZTO-doped hybrid electrolyte demonstrated the highest conductivity of 5 × 10^−4^ S cm^−1^, which appears to be seven times greater than that with no LLZTO. This resulted in the reaction between Li^+^ and LLZO fillers via acid-base interaction. Lithium salt dissociation will contribute to the carriers’ concentration, which increases ionic conductivity. Furthermore, the garnet ceramic filler acts also as a plasticizer in lowering the degree of polymer crystallinity, which improves the segmental motion of the polymer and eases the ions migration and so enhances the ionic conductivity. The morphological aspect of ceramic fillers such as particles, distribution of nanowire, and 3D framework in polymer composite electrolyte may impact the ionic conductivity. Amongst them, aligned nanowires when combined with polymers can offer an uninterrupted Li^+^ migration track. Figure 13 shows the morphological impact on the ion migration in solid polymer electrolytes.

It was found that a CPE doped with aligned inorganic nanowires demonstrates better conduction than non-aligned ceramic-filler-based solid polymer. This result was also confirmed by Cui and his co-workers [116,131]. They found that polymer electrolytes based on well-aligned inorganic nanowires (LLZO) reveal an ionic conductivity of 6.05 × 10^−5^ S cm^−1^ at RT, which is superior by one order of magnitude to that of nanoparticles or randomly aligned nanowire-based CPEs. The improved ionic conductivity results in the free motion of Li^+^ throughout the matrix with no crossing junctions on the nanowire surface.

##### Sulfide-Type Polymer Electrolytes

Sulfide-type PEs demonstrate high ionic conductivities of two orders of magnitude at RT [132]. Zao et al. developed a solid electrolyte membrane via the combination of thio-LISICON Li_10_GeP_2_S_12_ with PEO [133]. The electrolyte membrane exhibits 10^−5^ S cm^−1^ of ionic conductivity, which is greater than common PEO-based PE with an electrochemical stability window of 5.7V. Glass-doped glass-ceramic Li_2_S-P_2_S_5_ and ceramic thio-LISICON Li_4-x_Ge_1-x_P_x_ S_4_ (0 < x < 1) are the most suitable materials to enhance the performance of solid polymer electrolytes in lithium metal stability and offer additional choices in the selection of positive electrode materials.

##### Perovskite-Type Composite Polymer Electrolytes

Perovskite-type solid electrolytes Li_3x_La_2/3−x_TiO_3_ (LLTO) are well known for their interesting properties, such as their stability at high voltages. They exhibit a cubic crystal system with a space group of P4/mmm and C-mmm [134]. Recently, Zhu et al. investigated a PEO:LiClO_4_ polymer electrolyte with Li_0.33_La_0.557_TiO_3_ nanowires. The ionic conductivity of Li^+^ was found to be 2.4 × 10^−4^ S cm^−1^ at 25 °C [135]. The addition of LLTO nanowire to PAN achieved higher ionic conductivity than pristine PAN film. The composite electrolyte contains a 3D long-distance Li^+^ conduction network (3D), which reduces the detrimental impact of agglomerated inorganic ceramic nanoparticles in polymers [99]. This artificial 3D infiltration network with a large specific surface area provides a continuous Li^+^ transfer channel.

##### Fast-Ion Conductive Ceramic-Based SPEs

Fast-ion conductive ceramics show a high ionic conductivity of up to 10^−2^ S cm^−1^ at RT, but their poor contact at the interface limits their employment as solid electrolytes. The combination of polymer and fast-ion conductor ceramics can result in promising electrolytes for LIBs. Fast ion conductors are thermally stable and commonly composed of garnet-type, NASICON-type, and LISICON-type ceramics.

## 4. Summary

In this review, different types of electrolytes and their electrical and mechanical properties have been reported and studied to evaluate their effect on LIB performance. It was noticed that the electrolyte component and solvent in polymer electrolytes have a great influence on the ionic conductivity, Li^+^ migration, interfacial contact between electrolyte and electrode, mechanical properties, and the performance of the entire battery. The morphology of incorporated additive materials (nanoparticles, nanowires, nanofillers, salt, etc.) may well contribute to the amelioration of the ion transport pathway, which raises the lithium-ion conductivity. A basic understanding of the chemical reaction routes and the electrolyte structure would facilitate innovation in the battery. The structural, electrochemical, and mechanical properties of new promising materials should be investigated in advance for application in advanced lithium-ion batteries. The electrochemical behavior is inextricably related to the structure. IL-based solid polymer electrolytes appear as a promising material for long-term lithium-ion batteries despite showing low ionic conductivity but exhibiting more advantages than conventional carbonate electrolytes such as good safety, stability, good electrochemical performance, good mechanical stability, and enhanced energy density. Since solid electrolytes exhibit low ionic conductivity, ILs used in SPEs increased their conductivity. In a battery, porous materials appear to offer good properties in terms of lithium ionic conductivity, with no leakage and low interface resistance, and gel-based LIBs demonstrate a good working performance, long cycling life, and high energy density. However, commercial LIBs have been widely developed, and the use of liquid electrolytes in the battery showed some drawbacks, such as low electrochemical stability and poor safety, which limits their use in many fields. As an alternative, the development of solid CPEs has been found as a promising solution to replace liquid electrolytes because they are highly flexible, non-flammable, thermally stable, and highly safe. In this work, a basic comprehension of the mechanism of ion transport and interfaces for polymer-based composite electrolytes has also been explained. A summary of the recent investigations on CPEs, including polymer/ionic liquid, polymer/inert ceramics, polymer/MOFs, polymer/cellulose, polymer/garnet type, polymer/sulfide type, polymer/perovskite-type composite electrolytes, and polymer/fast ion conductive ceramics, has been reviewed. However, polymers combined with fast ion conductors compared to other SPEs have been largely commercialized because of their excellent conductivity and good interface contact.

For future development of promising efficient and low-cost solid electrolytes for battery application, the database of materials genome should be exploited in the analysis, characterization, and design of composite materials. Advanced methods of characterization should be also used to make a deep understanding of the material mechanisms, such as cryo-electron microscopy. This technique reveals the entire structure of atoms present in the battery materials and interfaces. The cryogenic TEM method appears to be one suitable candidate for a good interface analysis of solid electrolytes. Since the advanced research on the manufacturing of novel electrolytes for LIBs combines all the advantages, it still has to be studied in depth. Good polymer electrolytes need to be highly conductive, safe, highly mechanically and thermally stable, and easy for film formation.

## Figures and Tables

**Figure 1 polymers-14-03101-f001:**
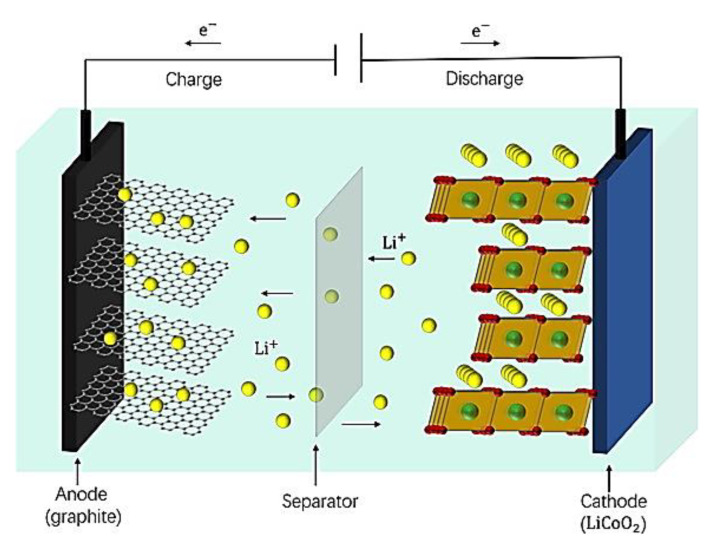
Schematic illustration of LIBs [30].

**Figure 2 polymers-14-03101-f002:**
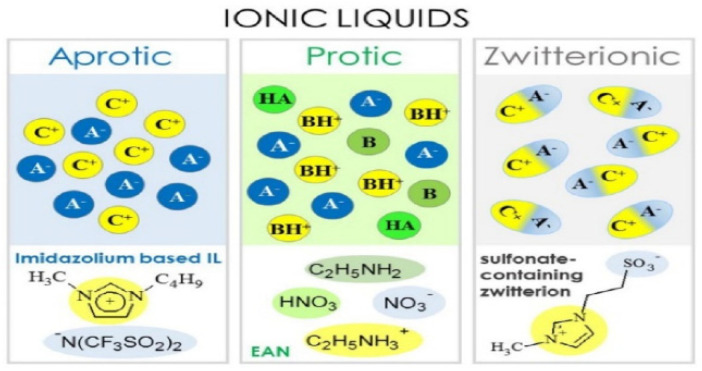
Classes of ionic liquids (aprotic, protic, and zwitterionic) (A^−^: anion, B: base, C^+^: cation, BH^+^: Brønsted base, HA^−^: Brønsted acid) [49].

**Figure 3 polymers-14-03101-f003:**
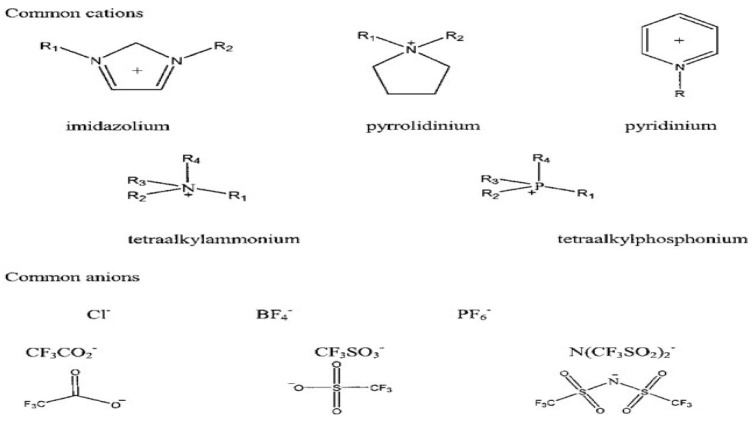
Structures of common cations and anions of ionic liquids [50].

**Figure 4 polymers-14-03101-f004:**
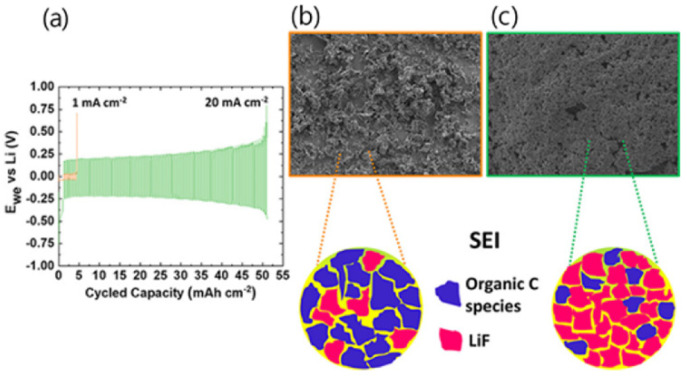
(**a**) Form of symmetric voltage for various current densities; (**b**) SEM picture of accumulated lithium at 1 mA cm^−2^; (**c**) micrograph of accumulated lithium at 20 mA cm^−2^ [68].

**Figure 5 polymers-14-03101-f005:**
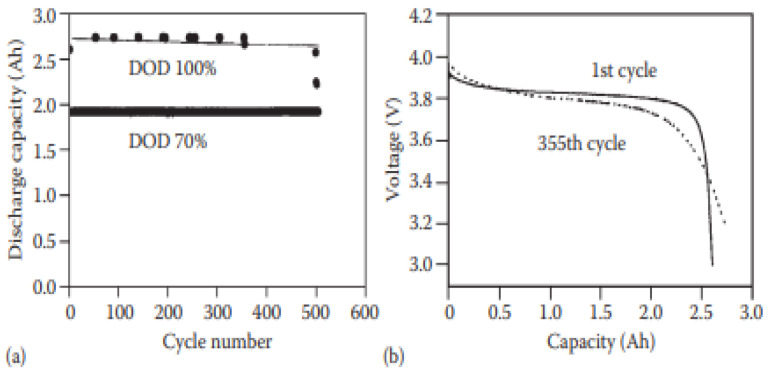
Cycling feature of a 10 Wh solid LIB: (**a**) at various drops of discharge and (**b**) discharge curves in different cycles. The LIB consists of LiCoO_2_ as the positive electrode and Li metal as the negative electrode; ethylene oxide and propylene oxide (EO–PO) copolymer are used as electrolyte [70].

**Figure 6 polymers-14-03101-f006:**
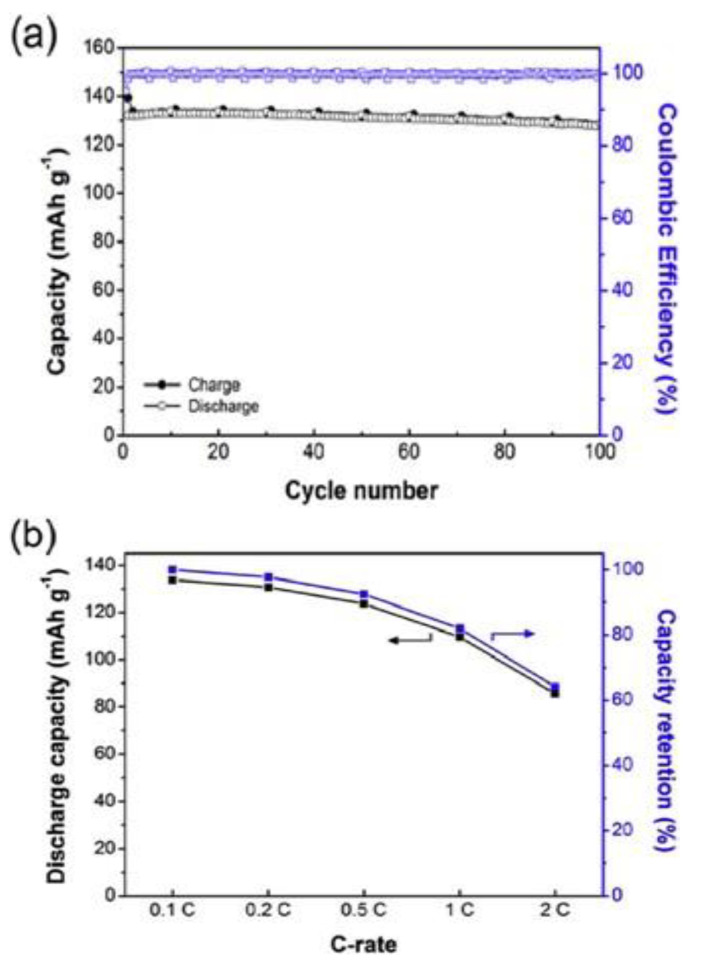
(**a**) Cycling performance of the solid electrolyte-based battery evaluated with a stable charge-discharge of 0.5 C at 30 °C and (**b**) capacity retention values of various discharge capacities [72].

**Figure 7 polymers-14-03101-f007:**
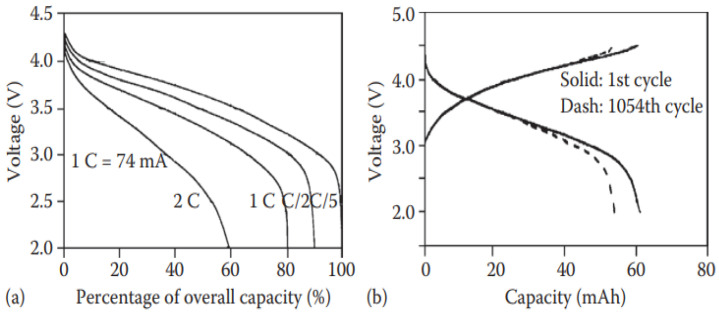
(**a**) Discharge curves for various rates and (**b**) charge/discharge curves in different cycles for the gel lithium-ion batteries (74 mAh) fabricated by Ultralife Co. Ltd [73].

**Figure 8 polymers-14-03101-f008:**
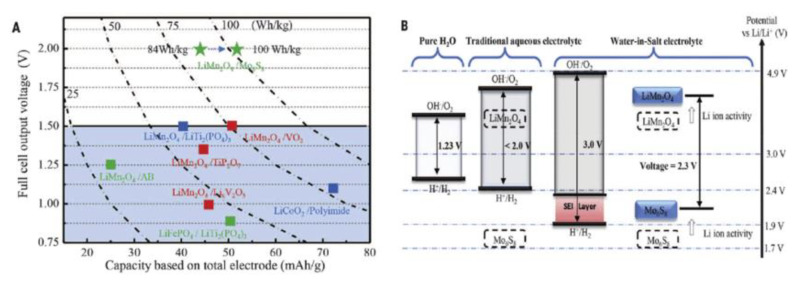
(**A**) Performance of each aqueous LIB based on different redox couples. Cycling stability: red color, 1000 cycles. (**B**) Representation of enlarged potential window for WiSEs together with the modulated redox couples of LiMn_2_O_4_ cathode and Mo_6_S_8_ anode due to high content of salt [66].

**Figure 9 polymers-14-03101-f009:**
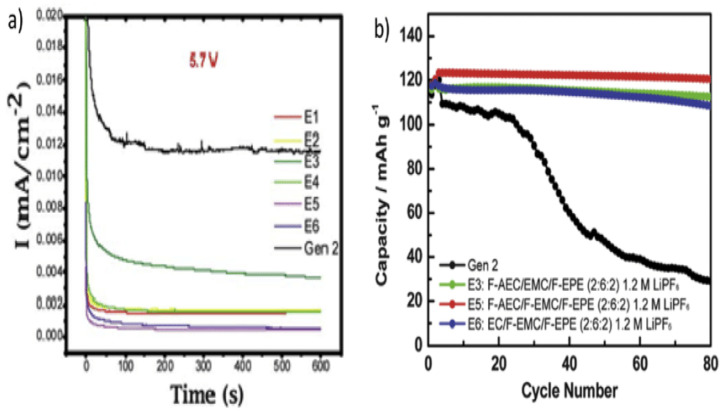
(**a**) Electrochemical potential window of Gen2 and fluorinated electrolytes E_1_–E_6_ at 5.7 V utilizing a traditional cell with three electrodes. (**b**) Cycling capacity retention of lithium titanate (Li_4_Ti_5_O_12_) (LTO)/LNMO cells with baseline electrolyte (Gen2) and fluorinated electrolytes E_3_, E_5,_ and E_6_ at 55 °C. Gen2: 1.2 M LiPF_6_ in EC/EMC (3:7); E_1_: EC/EMC/F-EPE (2:6:2) 1.2 M LiPF_6_; E2: EC/EMC/F-EPE (2:5:3) 1.2 M LiPF_6_; E3: F-AEC/EMC/F-EPE (2:6:2) 1.2 M LiPF_6_; E_4_: F-AEC/EC/EMC/F-EPE (1:1:6:2) 1.2 M LiPF_6_; E_5_: F-AEC/F-EMC/F-EPE (2:6:2) 1.2 M LiPF_6_; E3: EC/F-EMC/F-EPE (2:6:2) 1.2 M LiPF_6_ [83].

**Figure 10 polymers-14-03101-f010:**
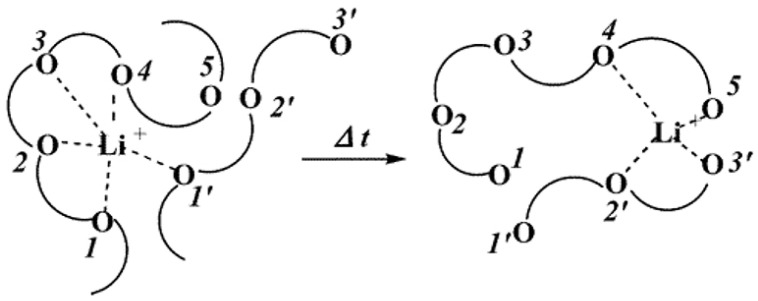
Structural representation of Li^+^ transport in PEO-doped polymer electrolyte (PE) [74].

**Figure 11 polymers-14-03101-f011:**
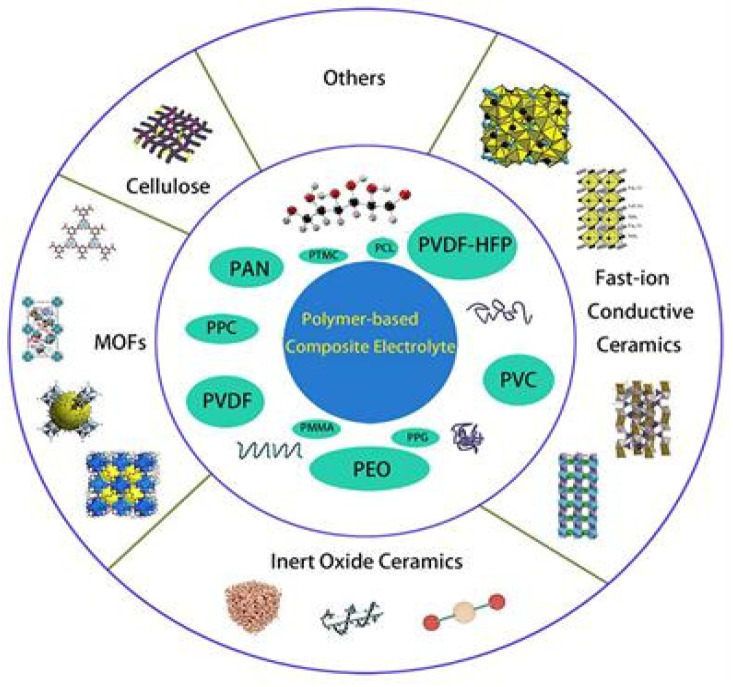
Classification of certain composite electrolytes doped with polymers [99].

**Figure 12 polymers-14-03101-f012:**
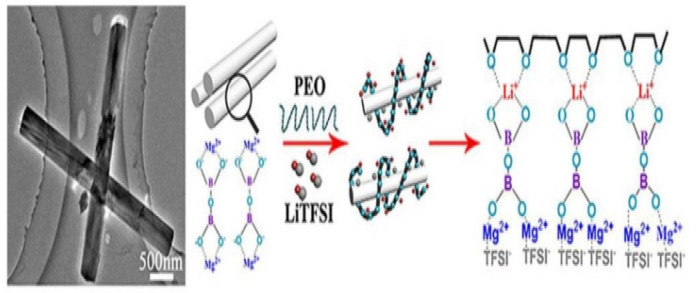
Schematic representation of Li-ion transfer in Mg_2_B_2_O_5_ improved based on solid CPEs [110].

**Figure 13 polymers-14-03101-f013:**
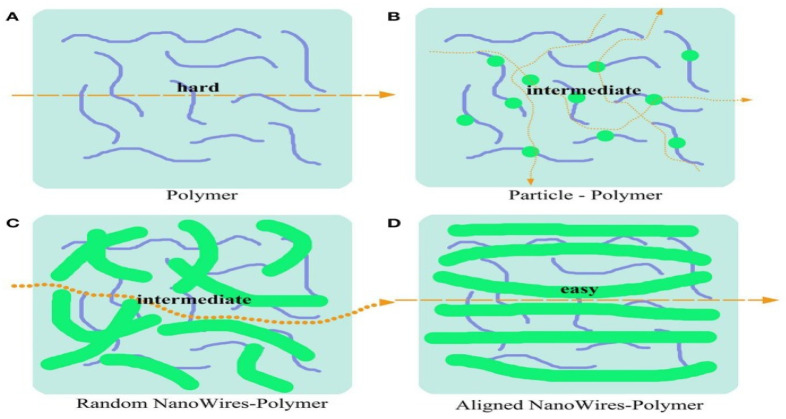
Schematic diagram of ion transport mechanism in SPEs incorporated with different varieties of ceramic fillers. (**A**) Ion transfer pathway in the pure polymer. (**B**–**D**) Ion transfer pathway in nanoparticle-polymer-based CPEs (**B**), random nanowires (**C**), and aligned nanowires (**D**) [99].

**Table 1 polymers-14-03101-t001:** Properties of some conventional organic solvents [83].

Solvent	Density d, g/cm^−3^ (25 °C)	Relative Permittivity ε_r_ (25 °C)	Viscosity η, mPa s (25 °C)	Homo Energy E_Homo_, eV	Lumo Energy E_Lumo_, eV	Melting Point mp, °C	Boiling Point b_p_, °C	Freezing Point f_p_, °C	Formula Weight (FW) g/mole
**EC**	1.32 (40 °C)	90 (40 °C)	1.9 (40 °C)	−12.86	1.51	36	238	143	88
**PC**	1.2	65	2.5	−12.72	1.52	−49	242	138	102
**DMC**	1.06	3.1	0.59	−12.85	1.88	5	90	17	90
**EMC**	1.01	3	0.65	−12.71	1.91	−53	108	23	104
**DEC**	0.97	2.8	0.75	−12.59	1.93	−74	127	25	118

## Data Availability

No new data were created or analyzed in this study.

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
