# Peer review of "The Impact of Polymer Electrolyte Properties on Lithium-Ion Batteries"

_polymers, 2022, doi:10.3390/polym14153101_

Round 1

Reviewer 1 Report

Based on the knowledge that this is a review paper, I believe that it is adequately written. It has good content as well as length. It's the right title. The introduction as well as the other chapters are adequately written. The results contain everything you need and each subtitle is accompanied by a satisfactory text. The summary contains the most important things and it is in line with the whole paper. I think that you should go through the work once again due to technical errors such as: excess or lack of space between two sentences, lack of space after the end of one and the beginning of another subtitle, etc.

Author Response

Based on the knowledge that this is a review paper, I believe that it is adequately written. It has good content as well as length. It's the right title. The introduction as well as the other chapters are adequately written. The results contain everything you need and each subtitle is accompanied by a satisfactory text. The summary contains the most important things and it is in line with the whole paper. I think that you should go through the work once again due to technical errors such as: excess or lack of space between two sentences, lack of space after the end of one and the beginning of another subtitle, etc. The corrections are highlited in yellow.

Reviewer 2 Report

Comments are given in pdf file

Author Response

Response to Reviewer 2 Comments

Peer review

Type of manuscript: Review

Title: The impact of polymer electrolyte properties on lithium-ion batteries

Authors: Azemtsop Manfo Theodore, Badi Nacer, Saleh A. Alghamdi Hatem A. Al-Aoh, Abderrahim Lakhouit, Pramod K. Singh, Mohd Nor Faiz Norrrahim, Gaurav Nath

Journal: Polymers

The review is timely and needed to provide scientific community with the information about the recent developments in the components of Li-ion batteries (LIBs). Thus, the submitted manuscript will be of interest to the readers of Polymers.

The authors should carry out major revisions to improve the quality and the language of the manuscript.

  1. The English language should be improved. There are missing articles (a and the), poor grammar and punctuation, problems with verbs, misuse or wrong use of English words (in the abstract, ‘In the last ten decades’ means in the last 100 years, which is misleading LIBs has much shorter history; ‘widows’ instead of ‘windows’, )

Response 1: Please provide your response for Point 1. (in red)

The authors have taken the reviewer’s suggestions into consideration and have made all the required corrections accordingly and other occurring mistakes in the manuscript.

Solid-state polymer electrolytes (SPEs), a specific class of polymer, are considered as promising candidates to replace current organic liquid electrolytes due to their mechanical properties and electrochemical stability against lithium metal [23].

Gel polymer electrolytes which contain liquid plasticizers have given much attention in the field of energy storage because of their specific characteristic, fairly high ionic conductivity, and enhanced interfacial charge transfer [24,25].

The chemistry of secondary battery batteries in aprotic electrolyte is very close to the chemistry of primary ones. There are a few recent studies that demonstrate that utilizing a liquid electrolyte in conjunction with a porous polymer structure can provide good transport properties and cycling performance in different configurations of lithium battery [31-35]. In the 1 M of lithium bis(trifluoromethanesulfonyl)imide (LiTFSI) with 1,3-dioxolane, Dimethoxyethane (DOL-DME) and 0.5 M LiPS (lithium polysulfide) electrolyte, the transference number was reported as 0.98, which may be the highest transference number ever reported for a gel polymer electrolyte containing free salt, along with 1.14 × 10−3 S cm−1 ionic conductivity[27]. Wang and colleagues used a similar crosslinked polymer as a coating on a polyethylene separator, in the presence of 1 M Lithium hexafluor phosphate (LiPF6) with ethylene carbonate-ethyl methyl carbonate-diethyl carbonate (EC-EMC-DEC) [34]. The transference number was 0.72 along with 10−4 S cm−1 ionic conductivity at room temperature. The impact of polyethylene dimethacrylate (PEGDMA) by addition of water in the prepolymer solution was studied. As a result, it was found that conventional Li-ion battery electrolyte contained in the porous PEGDMA network leads to augment the lithium transference number to 0.79 while maintaining the ionic conductivity in the order of 10−3 S cm−1 [27].         

  1. The Introduction is very generic – it could be expanded; more specific details should be The more specific data (including some values) should support the

statements/phrases such as: ‘high energy density’ (line 40), ‘The change in the content and physical and mechanical properties of the surface of active materials has drawn particular attention’ (lines 53-54); specify the nature of ‘gas produced’ (line 59); detail ‘many

drawbacks’ (line 58). A schematic of LIB with its main components and ions flows would be very useful following the line 45-46.

Response 2: Please provide your response for Point 1. (in red)

The authors have considered the reviewer’s suggestions and expanded the introduction part with more details are present and some specific data are included to support the statements/phrases: ‘high energy density’ (line 40), (lines 53-54); and line 59. The details about the drawbacks was given too0

Line 40. The energy density of the LIBs can commonly be improved by high voltage active cathode materials, anode materials, and electrodes. One of the main problems regarding high voltage cathodes and LIBs is the decomposition of the electrolyte at more than 4.2 V Li/Li+ [3-9]. Hence, to develop Li-ion conductivity, the stability of the material succeeding the electrolyte can also be considered as a crucial electrochemical parameter in the implementation of next-generation devices that hold higher cell potential.

Lines 53-54. For electroactive materials to be considered as latent candidates for LIBs, the immediate requirement is to obtain the reversible capacity, excellent ionic conductivity, good life span, excellent diffusion rate of lithium into active materials. The graphite permits the intercalation of only single Li with the 6 carbon atoms that result in the stoichiometry of LiC6, providing the equivalent reversible specific capacity of 372 mA h g−1. Therefore, it is urgent to replace graphite anodes on materials with higher capacity, energy, and power density. The modification of electrolyte is a well-accepted way to improve the performance of LIBs. Small amount of electrolyte additives incorporated is often introduced to develop the SEI layer at the interface between active material and electrolyte [19-22]. Electrolyte additive materials are often used in a small amount to create the SEI layer at the active material/electrolyte interface [23-26]. Solid-state polymer electrolytes (SPEs), a specific class of polymer, are considered as promising candidates to replace current organic liquid electrolytes due to their mechanical properties and electrochemical stability against lithium metal ­[27].

                 Specify the nature of ‘gas produced’ (line 59);

 Hydrogen fluoride

Detail ‘many drawbacks’ (line 58).

LIBs also suffer from many drawbacks such as, a poor rechargeability and susceptibility to misuses leading to fire even explosion.

A schematic of LIB with its main components and ions flows would be

very useful following the line 45-46.

Fig. 1. Schematic illustration of LIBs[30].

The transfer of lithium ions inside the battery is accompanied by the flow of charge in the external circuit, so the efficiency of lithium ion transfer in the electrolyte affects the capacity of the battery.

  1. It is incorrect to state that ‘solvent molecules react with the anode material (graphite)’ (line 48), the reaction would lead to a consumption of the electrode. Graphite does not react with electrolytes or Please correct.

Lithium ions moving along with solvent molecules, and passing through the separator, react with the anode material (graphite) at the electrolyte/electrode material contact and create the solid electrolyte interphase (SEI) layer.

  1. The abbreviations/acronyms should be explained after the first mention in the text. The authors do not follow this rule: IL (line 66) and NMR (line 89). Many abbreviations not explained at all: EMI, BMI, PAN, DCA, DOD, PP, EO-PO, PVDF-HFP, EC, DEC, SAPE, DMC, EMC, PC, DEC, LTO, TEM. The authors should check all acronyms and provide the readers with clear explanation what they

The authors have taken the reviewer’s suggestions into account and have provided their explanation of all the abbreviations in the new manuscript.

  1. The sections entitled Materials and Methods and Results are not applicable to the review. The authors do not describe their own materials and methods or results. Suggest to take these headings out and provide new

The authors have taken out the section of entitled Materials and Methods. As the authors have not used any new materials and methods in this work. They have investigated the different materials and techniques used in the work already published to show the impact of the outstanding electrolyte materials, and solvents and reported the properties of some electrolytes and results that could have an impact on the LiB performance. The authors have provided new subheadings as suggested.

  1. Does the phrase ‘dynamics of 1H, 19F, and 7Li,’ is related to atoms or Please specify.

1H, 19F, and 7Li,’ are related to atoms

  1. What does the phrase ‘the salt’s combination salts’ means? Clarify and correct in the

The authors have corrected the sentence.

 Nevertheless, the Li-ion transference number (t+) in the matrix was much higher, owing to the decrease in the coefficient ratios of TFSI/Li self-diffusion from 2 to about 1 moving from the salt combination to the polymer membrane

  1. In line 101, specify the names of ‘solid polymers’, take out ‘have’ from ‘with characteristically have high molecular weight’.

The authors have corrected and specified.

Examples of materials reported are solid‐state polymer electrolytes (SPEs) [40], with characteristically high molecular weight, gel electrolytes, or additive-based composite materials [41-45].

  1. Glass transition temperature is a characteristic of a polymer, so in line 105, phrase ‘ the electrolyte at T > Tg’ replace ‘electrolyte’ with ‘ a polymer’.

               The authors have corrected the phrase in the manuscript accordingly

  1. Chemical formula should be accurately presented in the text: the number of atoms (i.e., indices) must be given as subscripts, there should be no spaces between the elements, the ion charges must be shown as superscripts. I will scrutinise that each chemical formula is corrected and accurately presented in the revised manuscript. Some examples (not all the cases) of ill-presented formulae are in the lines: 116, 149, 151, 203, 311, 330, 331, 435, 475, 487, 489,

The authors have considered the reviewer’s suggestion and corrected all the subscripts of chemical elements in the manuscript.

  1. In section 2.2, give examples of flammable solvents (line 123) and non-flammable solvents (line 126).

The authors have provided examples of flammable solvents (line 123) and non-flammable solvents as suggested by the reviewer.

Additionally, electrolytes made with traditional organic solvents are highly flammable such as acetone, acetonitrile, acetic acid, benzene, acetic acid …etc, and volatile owing to the non-stability of these organic solvents for a large temperature range [36][49], which can conduct to the device’s degradation and conflagration. For this reason, the use of common non-organic solvents such as carbon tetrachloride, sulfur dioxide, sulfuryl chloride fluoride, dinitrogen tetroxide, antimony trichloride, and bromine trifluoride..etc with the potential possibilities has led to the manufacture of non-flammable electrolytes with electrochemical and thermal stabilities.

  1. Line 142 ‘tetra alkyl phosphonium’ is one word.

The authors have corrected it accordingly.

  1. Line 146, correct ‘halide’ to ‘halides:’

The authors have corrected it accordingly.

  1. Lines 148-149, please check and correct the name and formula 1,1-trifluoro-N- ((trifluoromethyl) sulfonyl)methanesulfonamide)[N(CF3SO2)2

The authors have corrected it accordingly.

         bis(trifluoromethanesulfonimide (N(CF3SO2)2),

  1. Line 160, ‘bis (fluorosulfonyl) imide’ is one

The authors have corrected it accordingly.

  1. Lines 169, 283 and 284 mention ‘rate’ after temperature (the units of degrees Celsius must be presented correctly). The rate should have units, e.g. degrees/min. If authors are not

referring to the speed of heating the word ‘rate’ should be removed, otherwise provide

correct units.

The authors have made the correction as suggested by the reviewer. The authors do not refer to the speed of heating but the rate of current in coulomb of charge (C) at a specific temperature. C is the coulomb and not temperature in (°C)

  1. Table 1 does not carry any useful information apart from acronyms. It does not summarise properties or applications of ionic liquids. Either take out or give acronyms at the start of the

The authors have taken out Table 1 as suggested by the reviewer as all the concerned data are already mentioned above.

  1. The units should be separated from the numerical values with a space. For example, in line 203 ‘10Wh’ should be replaced with ’10 Wh’. The authors should check the entire text to provide correct scientific notation

                 The authors have corrected scientific notation throughout the text.

  1. The relevant reference numbers are missing in the text: line 213 - Suk et al.; line 237 - Ultralife Co. Ltd.; line 247-Boz and his team; line 271 – after ‘…and the cross-linked-solid state polymer’; line 310 - Zhang et ; line 412 – after ‘polymer composite inorganic

ceramic electrolyte has been developed.’

The authors have completed the missing reference in the new manuscript accordingly.

]. Jungdon Suk et al.[74] developed highly conductive electrolytes with an ionic conductivity of 7.6 ×10-4 S cm-1 for solid-state LiBs.

[74] Suk, J, Lee, Y. H.; Kim, D.Y.; Cho, S. Y.; Kim, J. M.; Kang, Y. Semi-interpenetrating solid polymer electrolyte based on thiol-ene cross-linker for all-solid-state lithium batteries. Journal of Power Sources, 334, 2016, 154-161.

[75] Cao, C., Yuan, X., Wu, Y., & van Ree, T. (2015). Electrochemical Performance of Lithium-Ion Batteries. Electrochemical Energy Storage and Conversion, 507–524. doi:10.1201/b18427-16

[27]Boz, B.; Ford, H.O.; Salvadori, A.; Schaefer, J.L. Porous Polymer Gel Electrolytes Influence Lithium Transference Number and Cycling in Lithium-Ion Batteries. Electron. Mater. 2021, 2, 154–173. https://doi.org/10.3390/ electronicmat2020013

                  [83] Y. Li, B. Ravdel, B.L. Lucht, Electrochem. Solid State Lett. 13 (2010) A95-A97.

         [84]Z. Zhang, L. Hu, H. Wu, W. Weng, M. Koh, P.C. Redfern, L.A. Curtiss, K. Amine, Energy &   Environ. Sci. 6 (2013) 1806e1810.

         [94] Yao P, Yu H, Ding Z, Liu Y, Lu J, Lavorgna M, Wu J and Liu X (2019) Review on Polymer-Based Composite Electrolytes for Lithium Batteries. Front. Chem. 7:522. doi: 10.3389/fchem.2019.00522

  1. Some illustrations provide a reference source, while others do not. If the authors produced these images by themselves, it is acceptable. However, if the illustrations are taken from publications the references should be provided to the following Figures: fig. 6; 7; fig.8; fig.9; fig. 10; fig.12.

The authors have provided the missing reference of the some illustrations taken from publications as suggested.

[68]Li Q.; Ardebili H. Flexible thin-film battery based on solid-like ionic liquid-polymer electrolyte. J. Power Sources. 2016, 303, 17–21.

[75]Cao, C., Yuan, X., Wu, Y., & van Ree, T. (2015). Electrochemical Performance of Lithium-Ion Batteries. Electrochemical Energy Storage and Conversion, 507–524. doi:10.1201/b18427-16

[85] Li, Q., Chen, J., Fan, L., Kong, X., & Lu, Y. (2016). Progress in electrolytes for rechargeable Li-based batteries and beyond. Green Energy & Environment, 1(1), 18–42. doi:10.1016/j.gee.2016.04.006

[76]]Xu, K. Non-aqueous liquid electrolytes for lithium-based rechargeable batteries. Chem. Rev. 104(2004) 4303–4417.

[104]Yao P, Yu H, Ding Z, Liu Y, Lu J, Lavorgna M, Wu J and Liu X (2019) Review on Polymer-Based Composite Electrolytes for Lithium Batteries. Front. Chem. 7:522. doi: 10.3389/fchem.2019.00522

  1. Line 216, poly (ethylene glycol) is one word.

The authors have corrected the word in the manuscript.

  1. Line 266, please provide a chemical name to a methacrylic

The author has provided the chemical name of methacrylic polymer (PMMA).

  1. Lines 270 -217, the statement ‘The stability of the water molecules was assured by the high concentration of salt and the cross-linked-solid state polymer.’ is controversial. Do the authors imply that the water molecules are not stable without salts and cross-linked polymers? What does it mean ‘stability’? Please clarify and correct in the

The authors have taken out that sentence as it is controversial, as the water molecules remain stable without salts and any polymer associated.

  1. Use the same acronym LIB in the entire text – do not replace with LiB (reminds of the incorrect formula of Li and B).

The authors have corrected the acronym with LIB in the entire text as suggested by the reviewer.

  1. In Table 2, provide the names of each parameter given in a column (i.e. bp boiling point) with the corresponding units. Does d in second column means density? If so correct the

The authors have provided the name of each parameter and their corresponding units as well as the unit of the density ‘d’ as requested by the reviewer in the new manuscript.

  1. Use the same presentation of conductivity units either S/cm or S cm-1

The authors have corrected the conductivity unit in the whole manuscript by S cm-1 as suggested  the reviewer.

  1. The parameters used in the equations (1-3) should be provide with the corresponding SI units, e.g. T in K etc. Provide units for Faraday constant, carrier density, mobility, the activation energy, reference temperature and glass transition temperature, ionic

The authors have provided the unit of each parameter in equations 1-3 as suggested by the reviewer.

  1. Line 275 units missing for 5.

The authors have added the missing unit for 0.5 in the manuscript as requested by the reviewer in the manuscript.

  1. Line 429-430, correct ‘young’

The word young’ modulus has been corrected by young's modulus

  1. Line 450, formula LiN is

The formula has been corrected as the reviewer suggested.

LiTFSI

  1. Line 465, find alternative to the term ‘inoffensive’ – not sure what it means in a given

The authors have changed the word ‘inoffensive’ as suggested by the reviewer.

Round 2

Reviewer 2 Report

The scientific notation must be improved further. This relates to the presentation of chemical formulae, ion symbols/charges, and units. The unit should be separated from the numerical values with a space. 

English must be improved, the articles still missing in many places. 

Some minor corrections are needed (see file attached)

Author Response

Reviewer Response (Round 2)

We are thankful to reviewers for giving their valuable comments. The pointwise reply is coated below (In Green Colours) to avoid confusion.

We hope that revised version is acceptable for publication 

The authors have not corrected the term ‘react’ (see the point 3 form the first review round). In line 57, the authors state that ‘lithium ions react with the anode material (graphite)’. A chemical reaction is associated with the consumption of reactants, i.e., the consumption and disappearance of graphite (anode). I doubt this is happening in LIB. Thus, I ask the authors to: either find an alternative to the word ‘react’, or reformulate the statement, or take it out to avoid confusion. 

Reply: We have corrected the sentences and corrections are given in Green Fonts to avoid the confusion

The accurate presentation of chemical formulae of compounds and ions needs some further work. As stated in point 10 (first round), the indices must be shown as subscripts the ion charges as superscripts. In the modified manuscript, the chemical formulae/symbols must be corrected in lines: 140, 145, 146, 158, 266, 269, 324, 384, 405-406 (the chemical formula cannot be split, remove the spaces between elements), 561 (spaces between elements to be removed), lines 592-593 (the chemical formula shown in two lines (this is not correct scientific notation)), 597, 617.

Reply: We have taken special care about subscripts / superscripts and formulas all over manuscript (Pl see corrected Green Fonts)

Table 1 should be reformatted – in he modified manuscript it looks that mp column does not have any entries. 

Reply: Now Table 1 reformatted (Pl see corrected Table 1)

young’s modulus should be written with the capital Y: Young’s modulus, please correct

Reply: Now its corrected.
